# What the Ground Says . . .

**Rita Occhiuto**

LabVTP Ville-Territoire-Paysage, Faculty of Architecture, University of Liège, 4000 Liège, Belgium;
r.occhiuto@uliege.be

**Abstract:** Ground, as a body incised by natural and human actions (European Landscape Convention), carries "stories", going beyond quantitative values. As in a text, it holds the keys to understand what it covers or hides. In its thickness, it shelters "implicit projects". Understanding its complexity requires a physical and perceptual commitment, challenging the body in space: dimensions gradually forgotten by Environmental Sciences. As a "threshold" between visible and invisible, Underground-Built-Heritage represents the reverse of the emerged world: hollow space, both generator and mirror of open space (cities, landscapes). The focus is on physical and mental relationships between these two worlds. Past and present relationships emerge, allowing hypotheses to reconstitute collective memories, practices, knowledge, and values, which serve territorial development. The "Three Countries Park" is a place for cross-border experimentation to test how UBH can rebuild common links for fragmented environments. The cavities of a geo-park (planned) and the tangles of underground mining architecture are the fragments of a vocabulary whose meaning communities have to relearn. Built undergrounds will, thus, emerge from common stories that revive the imagination of populations who have lost all notion of belonging to a place. UBH will become a vector of new territorial coherence linking the physical and mental perceptions of people.

**Keywords:** UBH; perception; imageability; cultural landscape; open project; intangible values

## 1. Introduction

Air, water, and soil are territorial elements of which we still strive to measure mainly the quantities, in order to identify their states, their degradation, or even their rarefaction. The resulting feeling of loss of natural resources reinforces and amplifies the use of quantitative approaches, rejecting anything that appeals to the qualities of materials and environments, to a secondary or even accessory role.

Through studies of the territory in Wallonia (BE), we noticed that this posture is linked, on the one hand, to the inability to read and understand the complexity of the existing and, on the other hand, to the negation of relations between the components of territorial systems. However, the notion of relations or interrelationships is one of the major elements of the innovation carried by the European Landscape Convention.

The territories, characterized by their surface extensions, acquire the status of landscapes when we consider the interrelationships between human and natural actions from which they are both originating and carrying. These allow us to better understand the contributions of populations in the transformations of places. In addition, taking into account human perceptions gives the territory a cultural and sensitive depth that allows us to get closer to the environments to reveal local stories and experiences, as if the lands could tell us their story [1].

Today, the scarcity of the land resource is beginning to reorient the goals of development. In addition, we question again the underground. In the past, it was to extract raw materials or to bury different technical networks.

Today, we are mainly looking for spaces that we no longer find on the surface.

Is that a good reason to come back to reading and working on the underground?

The underground is much more than a resource of new surfaces to occupy. It has always represented another kind of space, the reverse of what is on the surface of the earth, an inverted world, darkness opposed to lights, an infinite resource, perceived as inexhaustible (hollow space in a matter of unimaginable extent), etc. For centuries, men have been digging galleries and building underground. The development of modern cities and their territories, from the 19th century, reinvented their geographies, by creating vast intertwining of various networks that served the emerged urban space, whose horizontal stratifications were made up of moving materials: circulation of both people and goods.

The underground represented a new value: the hidden side of a society that on the surface organized the ceaseless rhythm of flowing lives, invisible flows and movements at the service of the visible world. Thus, the underground gradually became a place of anthro-posage [2] or an environment to be exploited which, when the economic needs waned, was abandoned. In addition, while, on the surface, the rhythm of movement and exchange still continued, in many industrial undergrounds, time stopped.

When abandonment has set in, silence and oblivion took place. However, built structures still punctuate the viscera of this same ground that has been so exploited and which we keep endangering. Aware of the limits of a culture of excessive exploitation of the earth resource, we can act to finally recognize other values. Indeed, the ground and underground are materials that bear the "traces" of what has happened engraved in the thickness of their bodies. Today, therefore, we can also recognize their role as bearers of various writings, to be considered both as "vestige" and stimulators of multiple "memories", linked to human and natural actions (ELP).

Thus, to affirm that the underground is, at the same time, vestige, trace, and memory is an indispensable prerequisite to underline the cultural and identity significance that these materials carry with them. This approach will be used for the case study of the Three Countries Park (in progress), where the artefacts, which have yet to be inventoried and reused, are mainly mining galleries and cavities left by quarrying activities. The underground also preserves built artefacts whose qualities of unusual spaces, mazes, tunnels, chambers, canals, veins, and various hollows, still exist and express the intense interaction between human and natural.

Indeed, the sedimentations and their temporalities speak of slow dynamics of formation, disturbed by the violence of a human force which, while advancing "*like a rhizome, springs up at points, in the form of opportunistic resurgences which suck, dig and transform the riches of the sub- soil in order to immediately modify its environments and balances*" [3].

These subterranean artefacts [4], constituting the Underground Built Heritage (UBH), are a resource that requires special attention. Beyond designating a category of spaces to find, from which to distinguish the many existing types, they offer the opportunity to carry out more in-depth reading. These open up innovative fields of interpretation leading to the engagement of both body and mind.

This contribution introduces the cultural landscape as an approach proposed for the COST action "Underground Built Heritage as a Driver for Community Valorisation". This project aims primarily at the recognition of the values of underground built structures, linked to human activities. Through the crossing of competences, the COST project aims to make these artefacts known, to associate them with cultural practices and/or planning policies that can once again make them a "*motor for economic and cultural redeployment*". Different case studies and their comparisons will make it possible to pool knowledge and to renew territorial practices.

The approach we propose, based on an interregional case study, emanates, above all, from field observation approaches that combine architecture and nature. We will not present here specific results, as we are at the beginning of the study. However, we aim above all to explore the processes which have produced, used, and, finally, forgotten these artefacts over time. In our approach, the time factor is a major element for the interpretation of the heritage character of places. Indeed, the interest in the constitutive dynamics of

environments makes time a determining factor in implementing practices that are not limited to simple conservation measures, or even blocking or stopping the forces in action.

On the contrary, the posture we want to adopt is essentially prospective nature and characterized by the will to follow the development of spaces in their being constituted by environments in progress or in fieri, contexts whose compositions may remain open or not concluded.

For sites made up mainly of natural materials, this means leaving the environments under the action, controlled, or accompanied, of on-going natural forces, by avoiding any museification process. This posture is part of the philosophy defended by the European Landscape Convention (ELC), which specifies that landscape management should follow "forward looking" [5] trajectories. Thus, grasping the in-motion quality of places, allows us to study them mainly from their alive and relational character and, therefore, always in a phase of transformation.

Starting from this thought of space, we are interested, more specifically, in the physical and mental links that are woven between the underground worlds, often remaining invisible, and their emerged and visible parts. We will discuss the importance of reconnecting the mind with the materials that allow any inhabitant to become one with their living environment: to perceive, touch, and experience in order to appropriate the places. Thus, the patrimonial interest of UBH resides not on their re-functionalization but, especially, on their re-meaning and mental re-appropriation, a process which makes it possible to give forgotten spaces a new cycle of life [6].

## 2. Ground: Thick Material Keeping Memory

The ground presents itself as an expanse that welcomes or repels humans in its folds, its depressions, or its bursts and abrupt forms. The ground is often considered as a continuous surface where only architecture imposes verticality and rupture in a horizon tending to infinity [7]. This soil is comparable to a leaf, a thin layer of matter on which rise worlds developed in height. Life, humans, and commodities move incessantly in this type of smooth or striated context [8], often in the absence of depth or roots. However, the ground is a thick material that we mark by the step and that we continuously incise to cultivate, build, and inhabit.

It is a thick material that keeps trace and memory of all action that has taken place, and it is in progress. This is a living matter that must be taken care of and that we must learn to read again. The landscapes, comparable to open books, are marked by various writings of men and natures composing complex palimpsests, of which only the surfaces are visible. However, what do these expanses hide? What has happened in these places [9] and in the grounds we inhabit

These are the questions that must still be asked in order to stop functioning as objects or merchandise of exchange caught up in the speed and compression of time [10].

The long times of soils formation allow us to give a value to the concretions of materials which keep memory of each stage of their composition.

Layered, compressed, or fossilized components allow us to visualize and become aware of natural formation processes. In these materials, every mark, hollow, or crack speaks of a mechanical or chemical operation that produced the change.

In addition, if the ground behaves as a matter that tells, what would it say?

What is he hiding? What does it protect? What does it preserve?

Or what can he finally talk to us about?

From the signs visible on the surface, it is often difficult to recognize the relationship with the depths of the subsoil. To grasp them, it is necessary to redevelop the necessary knowledge and a gaze that digs, searches, and questions itself, making it possible to link the forms visible on the surface with what occupies the underside of the face of the earth. As with the plant world, an effort should be made to relate the configurations of buried root structures to the aerial development of the plants themselves.

In addition, this wise look could be extended to all the types of space encountered.

Listening to places, feeling them, perceiving them through all the senses allows the imagination to move and re-develop the ability to create mental images, defined by K. Lynch as Imageability [11].

Reactivating this gaze makes it possible to reacquire the ability to capture the clues and details of places, give meaning to words and to reacquire the awareness that each gesture made on the surface of the earth is an incision which alters the original state, and which leaves a trace. With this strong awareness, we can easily understand that, for each lived environment, there could be a description or a text that transcribes its characters to transmit them to others, with the aim of re-launching the production of imaginaries: writing, saying, telling, drawing, describing to stimulate tactile perception, without going through the sense of sight.

It is this type of imagination that emerges from Italo Calvino's pages on "Invisible Cities" [12]. He uses the words as brushstrokes to sketch portraits of cities, all with their own character and structure of their own. These descriptions take the form of embodied images in the minds of readers. They become material, colour-, thickness, and shape and send us back to known or simply mentally re-arranged atmospheres, as if we had to pre-visualize them before drawing and constructing them concretely.

The strength of the descriptions lies in their ability to evoke atmospheres already perceived through lived experience. In this return, the images are not neutral.

They are loaded with emotions and feelings that can bring up memory of smells, lights, sounds, and dimensions that allow us to appropriate the writing and the spaces that emerge from it.

Calling on the lived imagination is a major stake in landscape project approaches because, to reorient the actions to be carried out in a territory, all the forces in place must be made participating and invested again in a process of change that makes sense to all stakeholders. Since the inhabitants are also agents of change, as are other natural forces, their full involvement, both physical and perceptual, is of great importance for the success of the projects.

The mental capacity to reconstruct the lived space is an aptitude to be reconsidered in the regeneration efforts of UBH because it is mainly a question of artefacts not visible on the surface. Indeed, they are both difficult to perceive and very rich in elements that stimulate the imagination. Preserving them without betraying their original nature requires approaches that do not reduce them to empty boxes without soul and without history. This approach will be developed in the context of our case study (Three Countries Park) by means of a precise survey of the existing cavities and the stories associated with them. This data, which is currently being defined and spatially mapped, is the basic tool for initiating the practice of "living lab" (means and objective of the COST Action). At the center of the process will be people's awareness of their hold on the underground landscape and, consequently, their possibilities today to safeguard it and make it evolve.

Through a more attentive approach to the components and to the landscape times that characterize the territory, we can preserve them as exceptional artefacts telling of their composite state, both natural, because they are made up of soil resources, but also artificial because they are sculpted or excavated by the hand of men [13].

The dual view of subterranean artefacts derives, on one hand, from the principles of the ELC, which recognizes in any landscape the interaction between human and natural forces. On the other hand, underground cavities, before being classified as heritage, are analysed as the result of human work (excavation and construction). More specially, Alain Roger used the reference to artefact and artialisation in the 1990s to describe the dual origin of spaces built with natural materials.

They must first be made to reappear, to allow the fruit of these underground labours to surface again and emerge with all their contents of altered nature and lost culture. From this re-emergence, we can initiate a new phase of the project, starting by giving the UBHs the opportunity to talk to the inhabitants, that is, to return to play a role in the cultural evolution of the communities.

By reconstructing the links between the city above and the structures below, it is possible to find the logical connections that have been forgotten, or even lost. The re-establishment of mental links between human actions and "landscape artefacts" is a central point in landscape practices to overcome the so-called limit state of deterritorialization (G. Deleuze, F. Guattari 1972) of our societies.

We could, thus, recognize spaces whose meanings and relations to the emerged world are linked: inverted images, as in a mirror, labyrinthine entanglements, threadlike constructions, tiered networks sinking into infinite depths, amazing chambers and cavities, places of silence and/or lively flow of water, etc.

They are all unexpected environments that challenge the mind, arousing interest in the depths of the earth and the fragility of the ground.

*DRAW to Reveal, Understand, and Re-Interpret*

These rediscovered spatialities make it possible to understand the reasons for these operations of transformation of the earth, as well as the relations, conflictual or collaborative, established with the natural elements of the landscape already there. To re-appropriate these environments on sustainable basis, surveying or capturing in situ, physically and mentally, marks the first step towards the revitalization of spaces. However, to ensure its permanence, it is necessary to guarantee its translation into transmissible and proactive elements.

Drawing is a valuable tool for documenting, recording, and showing what is not visible. However, it is also an effective instrument for showing the spatial relationships which still link surface life with that of the depths.

Drawing, therefore, lends itself to knowledge by sketching the lines of continuity of the elements which constitute and structure living spaces. Thus, it makes it possible to visualize the movements and the cavities buried in the thicknesses of grounds of which it is difficult to perceive the extent, the density, and the intensity.

The in-situ capture allows us to experience it.

The mind and the physical allow it to be recorded.

The outline of the drawing allows fixing and visualizing what the body has perceived.

Thus, the representation serves to give visibility to invisible structures that are already there. Drawing the UBHs is an essential step in reacquiring awareness.

Drawing, acting as a medium, helps to reconstruct the mental image which, accompanied by the physical engagement of the body in space, can bring the process of knowledge to completion. Places, passing through perception, begin to exist again by interacting with the space-time of our real experiences. The completion of this experiential journey makes it possible to restore to the built underground cavities a new value opening up to other perspectives of the meanings of places.

Our hypothesis is based on the conviction that we act as geomorphic agents [14] and that, if we want to re-launch a new life cycle for the networks of constructed underground spaces, we must consider them as acted materials (moulted from the inside or fruits of the material's own transformations), testifying to the thickness and the experiences sealed in the depth of the place. Thus, the action, the practice of the places by their interior makes it possible to reacquire the mental image of them by inscribing the environments in the lived memory. The re-appropriation of the context from the reactivation of relations to the material constituting the environments makes it possible to reverse the visions of the future of UBH. These, as living places, could be freed from the role of simple commercial objects offered for tourist consumption. They could, on the other hand, become the links of new systems of awareness of the fragility of natural environments. In addition, instead of offering us the richness of the materials that we extracted there (the full), they would start to exist again by confronting us with the intensities of their hollow spaces (the void). Thus, by confronting bodies with varied tactile experiences, ranging from majestic rooms with striated walls, to propped tunnels, so unstable and narrow that they take our breath away, UBHs could talk to users until it reaches the deepest layers of their emotions.

The appeal to the perception makes it possible to direct redevelopment actions towards programs aimed at reactivating the sensitive sphere of users with the aim of awakening awareness and, thus, arousing new imaginations, supporting a sustainable projectuality.

The objective is to give back to the territory projects which on the surface can still benefit from the qualities of properties inscribed in the underground, but from new mental relationships to be built between the emerged world and its inverted image inscribed in the material of the ground. The intention to make UBH places of awakening and awareness, connecting firstly the inhabitants (and then the tourists) to their anchoring sites, joins the cultural dynamic developed by the Benetton Foundation through the program "Curare la terra" (To heal the earth), a cycle of activities to raise awareness among citizens.

These study days were *"on experiences and researches oriented towards 'cultivation', understood as a mental skill and a set of practices able of expressing the meaning of the relationship between man and places that belong to his existence"*. [The objective was to investigate from] *"clues, practices and accomplished experiences which express a new necessary mental condition and a diffused sense of civic responsibility which can manifest as a care and cultivation of the inhabited places"* [15].

This type of questioning about the existing involves taking into consideration the broken links between inhabitants and inhabited areas. In addition, by the call to rebuild a bond lived in situ, it serves above all to rebuild first the imagination, through the tactile stimulation of the body and the triggering of emotions, to then restore to each other capacities to project themselves in a possible future. This starts, above all, with the full and conscious involvement of subjects who are regaining their capacity to be new actors: individuals engaged in the space and in the body of sites already there.

This opens up another way of conceiving and promoting the design of territories. It is *"a mode of action which provokes a substantial adhesion to a world in the making to be recognized as landscape. An open theme, which looks far beyond the category of 'project', understood as manipulation of forms on the part of expert knowledge, often far removed from the real processes of change"* [16]. The principle of "open project", like the notion of "open work" of U. Eco's", is used to define a mode of design that is not limited to the execution of a finished work. It refers to a project that still leaves open the possibilities of interaction and mutation. The adaptability of this type of design is particularly appropriate for natural environments which are continuously modified by human action.

This type of approach, applied to the case of UBH, transforms them into a real vector of change because they provoke a process of re-appropriation which will stimulate new and diversified projects. This is the aim of the COST action [17]. The possibility of engaging local actors in this process can guarantee sustainable actions that not only can regenerate underground spaces, but that can, at the same time, implement innovative and more specific economic and cultural recovery initiatives to the treated territories. Indeed, a major objective of this program is to develop practices that allow for better interaction with local actors: the graphic tools will be used to fuel debates and interactions in the living labs to be organized.

## 3. The Territories of Experimentation

For the development of this COST action, our territory of investigation and experimentation concerns the valley of the Meuse in the crossing of the territories of the agglomeration of the city of Liège, in Wallonia (BE).

We will deal more specifically with the section downstream from the city, connecting the terraces of the Liège Meuse to the plains of the Netherlands. Here, the emerged lands, as well as the depths of the subsoil, lend themselves to innovative projects putting the landscape at the -centre of economic and cultural development policies.

This vast landscape is characterized by the fluctuating relationships between the lands and the unstable waters of a river that has acted over time as a means of connection and cultural mixing, but also as a barrier. The state of the land and its modifications is, therefore, a very significant subject for the populations of these regions.

The numerous excavation points and the presence of slag heaps (mountains of slag deposited on the surface of the land) punctuate the landscapes with significant clues. However, the passage of time has changed both the image and the memories of this territory.

It went from being a "Black Country" to a "Green Country". The accumulations of slag heaps, colonized by spontaneous vegetation, today punctuate a vast, richly hilly area and bear witness to a bygone past. Even the aquatic territory has changed. The plain spaces, occupied by variations in wandering water and floating lands, have left space for a long river line disciplined by dammed banks, doubled by flow regulation and river transport canals.

The landscapes, under the blows of a voracious industrialization, have been metamorphosed and the populations have participated in these changes by integrating the characteristics of the environments not only visually but also physically. This was true especially in the underground, through continuous excavation, guided by narrow coal seams where temperatures rose and air decreased as one went deeper into the ground.

Over time, only a few memories and built artefacts remain of the miners' experiences, the only visible emergences of which are the slags heaps or the *Belles Fleurs* when they have not been demolished.

In most cases, these artefacts are ignored, or even completely erased on the surface.

However, the cavities still exist. In addition, beyond the coal mines, the most remarkable underground sites are most often the cavities formed by underground quarries or by water intake works, especially in towns.

These places that witness the overexploitation of the ground are still there and ask to be put back in value or simply to exist.

In the Liège region, among the best maintained and exploited places, there are the cavities of the Montagne St. Pierre, located on the northern border with the Netherlands, on the edge of the Meuse and the site of the Blégny-Mine coal mine, located on the eastern plateau of the valley, towards the border with The Netherlands.

In both cases, re-use programs for tourist purposes have already emerged. Both participate in cross-border networks for economic and tourism development. However, that does not make these underground constructions exceptional places to match their experience, their history, and, above all, their implicit potential, still so little known and exploited to feed the culture of local communities, living on the surface, and increasingly disconnected with this past hidden in the ground.

In the Unesco Heritage Site of Blégny-Mine (Figure 1), there is an intense culturalpromotion program which feeds a cross-border tourist network.

One can experience the depth, the darkness, the sounds, and the working conditions of the past, through the visit of long narrow tunnels that touch the sensory sphere of visitors. Although the place is already rich in quality activities, it remains isolated and fails to fully assert itself as a real driver of development.

In the conurbation of Liège, the part of the territory made up of the bed of the Meuse in its course towards The Netherlands, and the terraces and plateaus located on the right bank, offers a set of rich and diversified landscapes which merge with the environments belonging to the agglomerations of Maastricht (to the North) and Aachen (to the East).

Here, the surface landscape blurs the tracks and contradicts the human will to divide the lands not only according to national or political status but also and, above all, by the invisible separations generated by linguistic differences. In this seemingly continuous piece of territory, the sounds of languages, such as Dutch, Flemish, German, and French, resonate.

Yet, this land is a unique artefact, and the most unifying elements that connect them are: air, water, ground, and underground. Here, the geographical and geological landscape is contradicted by the cultural landscape, where man has used languages and the outlines of land divisions to slice, differentiate, separate, and, finally, fragment what was the result of long-term work: the underground.

Since the end of the 20th century, this cross-border region has followed the equally innovative and rare logic of the concept of landscape as a vector of continuity and union.

The major objective has been to promote a country made up of the same coal substrate linking the emerged landscape underground.

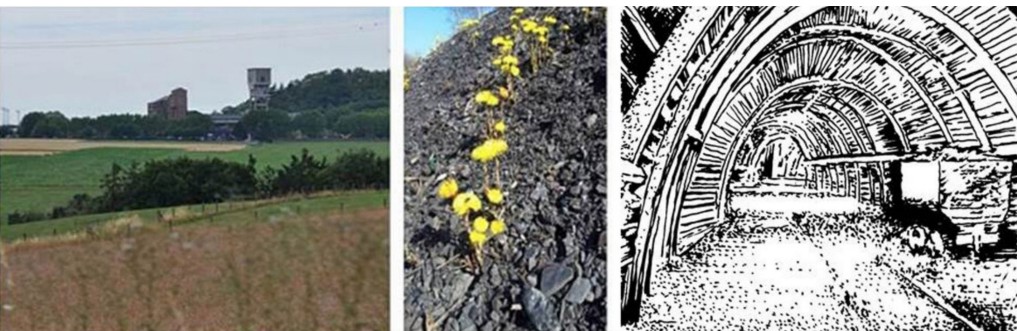

**Figure 1.** «Belles Fleurs» in the rural landscape of Blégny (BE), are the only signs witnessing on the surface the existence of mining activities. Outcrops of coal, hosting biodiversity, are indicator of new life cycle.

## 4. Discussion: The Three Countries Park

The selected case study and experiment concerns precisely this vast area, which spans three countries: Belgium, Germany, and The Netherlands. Since 1976, his territory has formed the Rhine-Meuse Euroregion, which concerns the agglomerations of three main cities, Liège (BE), Aachen (DE), and Maastricht (NL), and three regions, German-speaking Wallonia (BE), Westphalia -Rhenania (DE), and Limburg (NL). In 2008, by seizing the opportunity of the Interreg III and IV programs, the idea, and then the order of "*creating a new identity for a regional group without defining a precise territory and seeking new relationships between regions and cultures through the landscape*" [18], took shape.

The landscape emerges as a concept open enough to be shared.

Based on the existence of three urban centres, the principle of "Green Metropolis" or "Grünmetropole" is adopted, on a German political proposal.

The Agence Ter, led by the landscape architect Henri Bava, won the competition whose objective was to carry out a Masterplan for the renewal of the identity of these post-industrial territories. The landscape remains the principle with the necessary coherence for the development of the project.

Based on observation and fine in situ reconnaissance methods, this landscape project has been designed for a territory without borders, limits, thresholds or entrances. The continuum of the lived and travelled landscape stands out as a strong characterization element of the territory. Reflection on the concept of continuity and extent leads to the notion of horizon, which refers both to "*visual horizons and to invisible or subterranean horizons: soil thicknesses, geological layers*" [18].

The landscape designers were impressed by the strength of the continuous underground layer of coal crossing the borders (200 km long and 30 km wide) and made it the basis of the project and of all the dialogue and consultation actions carried out in the area. Although invisible, the underground coal seam, present in local collective memories, stands out as the element symbolizing both the past and the future of the cross-border project. This concerns two major mobility systems, one road and the other cycle, crossing the Euroregion and bringing together a series of specific points to be achieved over long periods of time. Thus, an open project takes shape, based on different temporalities and on the idea that, from a material and concrete substratum, can arise multiple actions and places on the surface linked by intangible connections.

Once again, from a strong and continuous landscape base, the landscape project acts by identifying local specificities to be linked through connections of meaning, or through stories supported by collective memories. The concrete landscape is there, but it is invisible. The landscape to come, the one to be built through the project, is invisible. It will emerge only in points, where there is the collective commitment of its inhabitants.

Thus, the project is not formal, aestheticizing, nor fixed. It is intangible, flexible, and evolving, as it reorients itself according to the opportunities built up over time.

The conceptual innovation of the Agence Ter proposition involves the use of the metaphor of DNA, illustrated by a diagram showing the intertwining of the two structuring pathways. The double helix represents the capacity for regeneration and movement. It also refers to the qualities of DNA: intelligence and complexity.

These are transposed to the different scales of the territory. The metaphor seems particularly well chosen because, by transmitting the idea of rebirth through action, it manages to communicate a positive and proactive image through a project that enhances both landscapes and inhabitants heavily affected by industrial decline.

The idea of an intangible territory to be accompanied over time by continuous practices that inaugurate and fuel the *open project* gradually takes shape.

Then, this dynamic of continuous project could be supported by other research actions. From the objective of building territorial cohesion, the ESPON project "3LP3LP–Landscape Policy for Three Countries Park" was born.

For this project, too, the "*Landscape serves as common denominator to reconcile different levels, competences and interests of stakeholders*", in order "*to create a shared vision integrating various spatial functions and responding to European challenges*" Thus, "*the project's objectives became:*

1. *The examination of the 3LP's European identity and dynamics based on previous ESPON studies*
2. *The design of a spatial landscape perspective for the future development of the 3LP.*
3. *The development of policy recommendations at the interface between the 3LP landscape perspective and EU policy*" [18].

The research highlighted the existence of landscape qualities based on typologies specific to diffuse peri-urban fabrics where nature and culture components are intertwined and supported both by the perceptions of the inhabitants and by the interests of local development actors.

"*The landscape perspective is meant to be a vision document with no formal or binding status. It has been appreciated by the stakeholders and is now used by the 3LP partners with spatial and landscape planning competence as well as many projects and local organizations as a source of reference, guidance and inspiration*" [18].

The Three Countries Park remains a vast area with blurred boundaries and the capacity to vary, adapt. and transform according to natural (climate change, risks, resources, etc.) and human needs (demographic growth, density, etc.) as if it were a living organism in which the communities are involved in preserving its balance and promoting its evolutions (Figure 2).

The Three Countries Park is more than an area. It is also a platform for any new initiative ensuring the continuity of the project. The aim is always to promote the economic and cultural development of a territory, preserving its characters and retracing its identities.

Currently, several cultural consultation actions are being carried out. From the objective of making the underground coal layer the main unifying principle of the open territory of the Three Countries Park, the theme of the value of the UBH spaces constitutes an opportunity to continue the project. Moreover, the deep landscape and situated nature of this theme makes it possible to reinforce the initial principle of rebuilding a strong and diversified identity that can transform, into a positive experience, the tangible and intangible cultural baggage of these lands incised and upset by industrialization.

The built undergrounds can once again highlight the unifying value of the landscape, offering inverted images of this cultivated territory. The built underground structures can be distinguished into two main categories: mining galleries and quarries.

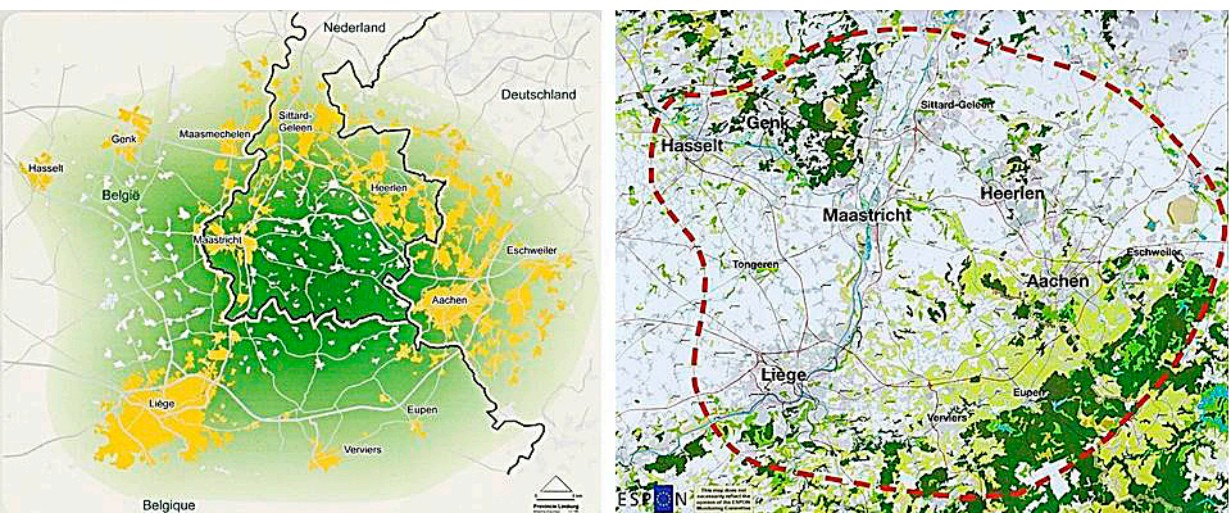

**Figure 2.** Extension of the cross-border territory of the Three Countries Park. Source: LP3LP Book, "Landscape Policy for the Three Countries Park", 2014 ESPON, The Three Countries Park. Authors and designers.

The mining sites are numerous, and they punctuate the Euroregion. They require more in-depth studies concerning the links they maintain with the urbanized fabrics on the surface. Mapping their underground expanses reveals aspects that are still unexpected and hidden. Revealing the root systems of certain farms through drawing would make it possible to reconstruct lost connections or to sketch new ones, opening up relationships to be reconstituted with the surface.

Reestablishing these links would also make it possible to rediscover the reasons for urban diffusion, as it is highly dependent on soil resources. For this reason, dispersion can be considered a phenomenon produced by the landscape itself. This contradicts the simplistic view of urban advancement as the main factor of landscape destruction. The work to be developed on the basis of this underground typology aims at a reversal of vision allowing to enhance the existing through a new look at the contributions and opportunities that these labyrinthine environments offer to local communities. Above all, this would reduce the gap that has widened between the new generations and the generic areas that they are occupying. These have no longer the status of landscape, but that of land for undifferentiated occupation.

The second category of spaces is made up of different types of cavities dug in rocks to extract construction materials. These spaces are labyrinthine, too. In most cases, they appear as sequences of pieces with differently marked walls, striated, or covered with drawings or writing. Spaces of more spectacular dimensions are reused for events, shows, or dinners. There is no lack of imagination in the reinterpretation of spaces. However, these activities remain punctual and do not yet constitute network actions.

The research that we are developing by the Action COST on UBH meets the objectives of the Three Countries Park and can intensify the cultural significance of the departure. This common capital constitutes a very strong starting point for strengthening identities while building transnational solidarity. In this extended landscape and with multiple open horizons, the underground spaces are, above all: ancient underground flint stone mines; natural karstic phenomena; limestone quarries; and lead, zinc, and coal mines. To these spaces can be associated: visible artefacts, such as: heaps; landmarks; architectural specific typology; and special biodiversity.

The cavities produced by the extraction of stony materials form the basis of a more specific network that defends the identity of the subsoil. The objective is to create networks of underground spaces that would bear witness to the common geological features and exploitation methods that still link these cross-border regions. (Figure 3).

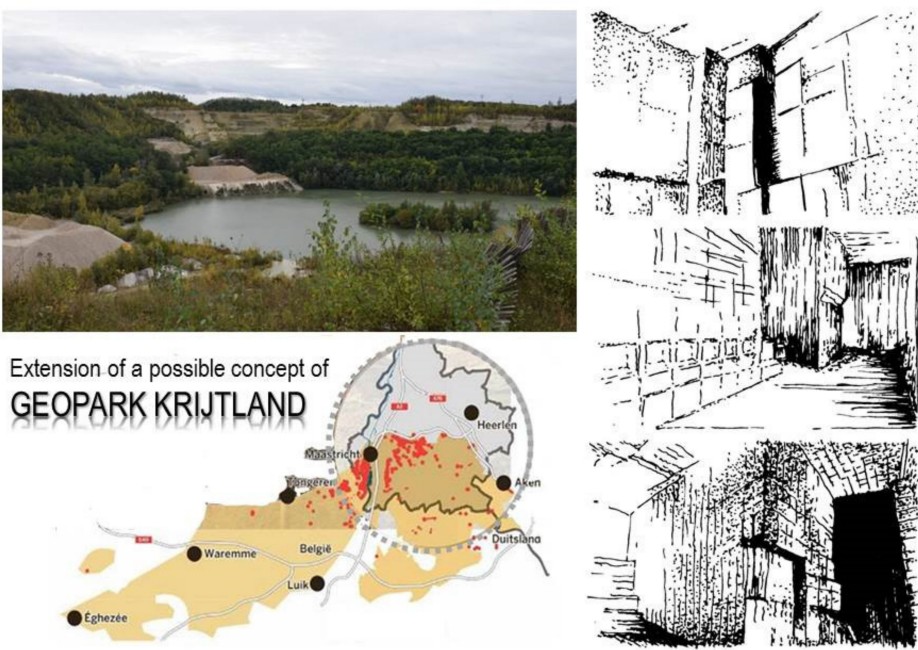

**Figure 3.** Quarries, the visible side of an underground unexpected invisible network of cavities for the project of a possible geopark. Source: De Limburger.nl 24 October 2018; Drawings and Picture by the author.

This one, starting from the same principle of the coal layer, proposes an underground which constitutes another kind of geographical map connecting the places marked by this type of material. The interest is to see these different geological layers cohabiting in a project their superimpositions, intersections, and articulations can give rise to various layered projects able to immerse the inhabitants in real underground landscape explorations.

However, the major objectives are constituted by the ideas, desires, and visions that these environments and their representations can still manage to emerge. This condition for designing is necessary to stimulate the creation of new entrepreneurial forms to be promoted in the whole extension of the territory. The transnational dimension, supported by the sliding or erasing of all kinds of limits, would favour cooperation centred on these artefacts and would focus actions on the value of work to be reconsidered as a factor of local pride and cultural distinction.

By resurfacing, underground spaces could suggest other types of economic activities and new trades connected with a profound revision of the value of land and its evocative power.

## 5. Conclusions

The study of this territory from this other angle of sensitive perception of the territory makes it possible to complete the knowledge of the environments. In addition, it will make it possible to work with local actors to develop aspects of land use planning that put identities and human capacities at the centre of the project to further question their perceptual sphere. From this reactivation derive more interactive methods of communication and sharing based on criteria for evaluating spaces that are both subjective (because individual) and universal (because they question space through the body).

The UBHs give back to the territory the opportunity to make all the parts of the inhabited and exploited geographies speak by returning to the geographical contexts the historical and temporal dimensions. These new networks will allow for complementary development themes to be articulated and will give the possibility of showing the meaning and the potentials of building solid intentional frameworks supporting projects whose writing can be continued. This methodology of investigation requires fieldwork but also long period of time and the iterative and cyclical involvement of local actors.

Observational drawing is an essential medium for taking in in depth notes, bringing out the invisible, the unspeakable, or the informal, and, above all, encouraging sharing and imagination, supporting the return to the desire to formulate wishes to be translated into future space projects. These monitoring methods are also a support for the livings labs to be built cyclically to accompany and to re-orient the projects.

**Funding:** This research received no external funding.

**Institutional Review Board Statement:** Not applicable.

**Informed Consent Statement:** Not applicable.

**Conflicts of Interest:** The authors declare no conflict of interest.

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
