# Peer review of "What the Ground Says…"

_sustainability, doi:10.3390/su132313420_

Round 1

Reviewer 1 Report

There is no clearly explained purpose or methodology in the article. These lacks impact the text's structure that combines theoretical considerations about the underground built heritage with an example of "the Three Countries Park" on the border between Belgium, the Netherlands and Germany. In my opinion, this original structure of the article should be clearly explained in the introduction. Otherwise, it is an essay rather than a scientific text. 

The first part of the manuscript has very emotional, ideological and even colloquial character. The phrases from lines: 62-66, 122-125, 203-206 should have never been included in the journal's paper. When the author mentioned "interpretation", "in-depth reading", "re-discovering", etc., she should propose adequate explanations of terminology. The author de facto focused on the underground sites made up mainly of natural materials but, reading the text, we would not find out why she avoided archaeological remains from such a rich historical territory? The author's sentence may illustrate this problem: "Long formation times of soils allow us to give a value to the concretions of materials which keep the memory of each stage of their composition". It's true – but we do not find in the whole text any notes about materials that are keeping the memory of human activity from the Antiquity or the Middle Ages. The rural and semi-rural landscape, the changes in soil or deforesting are the results of historical human activity which affected directly on the ground. What is more, recent archaeobotanical research allows reconstructing the natural history or modelling historical climate changes. The ground says about it also…

The author mentioned several times artefacts and their dual nature (lines: 170-172). According to Heidegger, the ordinary stone taken from the ground and hewed might be considered as a piece of art. Meanwhile, the aspects of the art theory were utterly neglected, except the technical and perceptual role of drawing (lines: 188-265).

Some author's statements sound too strong, and they should be proved, e.g. lines 178-179, 262-265. Also, some parts of the project description (lines 377-382) sound like a manifesto, not a neutral analysis. The definition of two categories of spaces (lines: 428-452) should be presented earlier. Finally, the conclusion should be supported by the result of the research presented in the article.

Editorial gaps:

The lack of references for the quoted text/phrases in lines 99-100, 337-338, 406-409.

References no. 13, 14, 15 do not need to be repeated one after another. 

Figures 1, 2 and 3 are posters, and they do not contribute any essential information for the text. I would instead prefer to see the map of mines and quarries from the Three Countries Park or the diagram with the systematic qualification of different types of underground heritage.

The source for figure 4 is needed.

I do not see the sense of two last sentences in lines 373-374. 

Author Response

Response to Reviewer 1 Comments

POINT 1

The first part of the manuscript has very emotional, ideologicaland even colloquial character. The phrases from lines: 62-66,122-125, 203-206 should have never been included in thejournal's paper. When the author mentioned "interpretation", "in-depth reading", "re-discovering", etc., she should propose adequate explanations of terminology.

The parts of the text considered too subjective have been modified to shed light, where possible, on the importance of referring to the soil as a living material, to be considered not as an "object to be transformed and exploited" (to the point of exhaustion), but as a "living element", a notion that is very present today in the landscape design disciplines. 
Clarification is provided in the introduction and in the part introducing the landscape approach to soil.It should be noted that the study to be carried out in the subsoil concerns all types of cavities and not only archaeological spaces. Of course, the mining galleries will be part of the archaeological contexts of this territory, but we are referring to "all" the modifications made by humans in the subsoil contexts. This is for :
1. try to identify spaces that are not already classified;
2. to give importance to human work.

POINT 2

The author mentioned several times artefacts and their dual nature (lines: 170-172). According to Heidegger, the ordinary stone taken from the ground and hewed might be considered as a piece of art. Meanwhile, the aspects of the art theory were utterly neglected, except the technical and perceptual role of drawing (lines: 188-265).

The notion of artefact is fundamental for the landscape because, for several authors such as Alain Roger et all, the contexts we live in are the result of a double action, due to natural agents (G.Vögt) and to humans (principles of the European Landscape Convention also.  Artefacts and duality of actions have been better explained now in the text

POINT 3

Some author's statements sound too strong, and they should beproved, e.g. lines 178-179, 262-265. Also, some parts of the project description (lines 377-382) sound like a manifesto, not aneutral analysis. The definition of two categories of spaces (lines: 428-452) should be presented earlier. Finally, theconclusion should be supported by the result of the research presented in the article.

The addition of legends for the figures and the addition of parts of the text makes it possible to make more explicit those parts of the text that are not "statements", but the reiteration of the intentions of the original landscape project for the Three Countries Park, a case study based on the Ter Agency's territorial project.  The lines in the "discussion" section summaries the intentions of this project.

Descriptive parts of this project have already been introduced in the introduction of the text, as advised.

Impossible for the moment to introduce more tangible results as the study has just started

First proofreading of English completed. Proofreading by a native English speaker is underway.

Reviewer 2 Report

As an archaeologist by formation, I particularly enjoyed reading that paper. The concept of UBH is new to me and it is an interesting one. I see a lot of possible applications and the paper I reviewed represent an informative and well-structured case study.

Author Response

Response to Reviewer 2 Comments

I would like to thank you for these comments. Indeed, we have an interest in the underground, as a living material.

By rediscovering the deep meaning of the life cycles that are inscribed in it, we intend to bring back attitudes that have characterised the work of human societies each time they have settled in a place.

Recalling this knowledge will help us to develop living labs from which we can attempt to reconstruct more holistic intervention hypotheses for the sites, with the collaboration of various specialists and local operators.  This is the basis of a methodology which refers to the landscape project and which focuses on a closer relationship with the natural and human materials present and interacting continuously.

Reviewer 3 Report

The author gives an interesting interpretation of the UBH topic by opening up related issues concerning the landscape. The text is clear since the conclusion where some arguments need to be specified and final statements supported by theoretical references.

In the complex, the contribution would deserve to be supported by a better figurative apparatus, especially considering the fact that the author refers to representation as a way to trigger a deeper comprehension of landscape relationship between under and over ground landscapes.

For specific comments on the text, see the attached file.

Author Response

Response to Reviewer 3 Comments

All comments that appear in the pdf version have been revised.

This concerns the better description of the objectives of the current COST project and the case study.

The images have been supplemented with captions that explain their figurative components. The expectation of a more comprehensive representation, illustrating the purpose of the article, cannot yet be fulfilled because:

  1. the research project is in its early stages and the first part will focus on surveying the locations on their more precise mapping;
  2. the type of schemes or spatial drawings will have to be completed by a graphic material which will make it possible to question the "visual images" of the inhabitants via a living lab to be carried out in September-October 2022, within the framework of the Cost Action UBH.
  3. For the moment, the Ter Agency's landscape masterplan, as well as the environments that will be the subject of a new network for the geopark, are sources that we cannot publish. The study will be an opportunity to produce images that better explain the interconnections between underground places, not yet considered connected.

From these documents, the experiment will focus on observing the reactions that these spatial images will provoke among the inhabitants and the various local actors. The more theoretical explanations are corrected in the text and in the notes.

First proof reading of English completed. Proof reading by a native English speaker is underway

Round 2

Reviewer 1 Report

The changes and additions made to the text have certainly contributed to its greater clarity.

The author has emphasised that she treated the soil as "a living material". However, it should be noted that apart from the European Landscape Convention of 2000, we also have the European Convention for the Protection of the Archaeological Heritage of 1992. Because the sustainable development paradigm contains the protection of heritage - whether in the forms of landscape (natural and transformed by human activity) and monuments or artefacts - it is worthwhile to be aware of the limitations resulting from objects' scale or their history. Moreover, some forms of heritage protection (e.g. exposition zones) result in "freezing" the landscape. May we talk about "a living material" then? The author's style of narration (rhetorical questions, repetitions) is not convincing to me.

However, I am sure this polemic is beyond the scope of the review. Therefore, after the author's explanations, I recognise that more changes in the manuscript are not necessary.

Minor fixes remain:
-    Line 483: Based on observation and fine in situ reconnaissance methods in situ…(?)
-    lack of publication in reference no. 4 (is it similar to no. 5?)
-    some citations are written in italic; the others are in quotation marks – The author should unify them. Moreover, references to each citation should be added (lines: 130-131; 314; 445-446)
-    there is no reference 5bis (line: 146). Consequently, we do not need to use "ibidem" if the pagination is the same.

Author Response

Thank you for your comments and suggestions. 
Heritage elements will be taken into account in the study to ensure continuity in local cultures. We have revised the notes and the italicised parts as requested.
